# Ten years of experience with endometrial cancer treatment in a single Brazilian institution: Patient characteristics and outcomes

**Cristina Anton** [1][◉][*], **Rodolpho Truffa Kleine** [1][‡], **Eric Mayerhoff** [1][‡], **Maria del Pilar Esteves Diz** [2][‡], **Daniela de Freitas** [2][‡], **Heloisa de Andrade Carvalho** [2][‡], **João Paulo Mancusi de Carvalho** [1][‡], **Alexandre Silva e Silva** [1][‡], **Maria Luiza Nogueira Dias Genta** [1][‡], **André Lopes de Faria e Silva** [3][‡], **Rafael Calil Salim** [4][‡], **Andrea Aranha** [2][‡], **Rossana Veronica Mendoza Lopez** [5][‡], **Filomena Marino Carvalho** [6][◉], **Edmund Chada Baracat** [7][‡], **Jesus Paula Carvalho** [1,7][◉]

1 Gynecologic Oncology Team, Instituto do Cancer do Estado de Sao Paulo ICESP, Faculdade de Medicina, Universidade de Sao Paulo, Sao Paulo, SP, Brazil, 2 Department of Radiology and Oncology, Instituto do Cancer do Estado de Sao Paulo ICESP, Faculdade de Medicina, Universidade de Sao Paulo, Sao Paulo, SP, Brazil, 3 Department of Digestive Surgery, Instituto do Cancer do Estado de Sao Paulo ICESP, Faculdade de Medicina, Universidade de Sao Paulo, Sao Paulo, SP, Brazil, 4 Laboratory of Surgical Pathology, Instituto do Cancer do Estado de Sao Paulo ICESP, Faculdade de Medicina, Universidade de Sao Paulo, Sao Paulo, SP, Brazil, 5 Oncology Translational Research Center, Instituto do Cancer do Estado de Sao Paulo ICESP, Faculdade de Medicina, Universidade de Sao Paulo, Sao Paulo, SP, Brazil, 6 Department of Pathology, Faculdade de Medicina FMUSP, Universidade de Sao Paulo, Sao Paulo, SP, Brazil, 7 Division of Gynecology, Faculdade de Medicina FMUSP, Universidade de Sao Paulo, Sao Paulo, SP, Brazil

◉ These authors contributed equally to this work.
‡ These authors also contributed equally to this work.
* cristinaanton@terra.com.br

**Data Availability Statement:** All relevant data are within the paper and its Supporting Information files.

## Abstract

Few reports have described the clinical and prognostic characteristics of endometrial cancer, which is increasing worldwide, in large patient series in Brazil. Our objective was to analyze the clinicopathological characteristics, prognostic factors, and outcomes of patients with endometrial cancer treated and followed at a tertiary Brazilian institution over a 10-year period.This retrospective study included 703 patients diagnosed with endometrial cancer who were treated at a public academic tertiary hospital between 2008 and 2018. The following parameters were analyzed: age at diagnosis, race, body mass index, serum CA125 level before treatment; histological type and grade, and surgical stage. Outcomes were reported relative to histological type, surgical staging, serum CA125, lymph-vascular space involvement (LVSI), and lymph-node metastasis.

The median patient age at diagnosis was 63 (range, 27–93) years (6.4% were <50 years). Minimally invasive surgeries were performed in 523 patients (74.4%). Regarding histological grade, 468 patients (66.5%) had low-grade endometrioid histology and 449 patients (63.9%) had stage I tumors. Tumors exceeded 2.0 cm in 601 patients (85.5%). Lymphadenectomy was performed in 551 cases (78.4%). LVSI was present in 208 of the patients' tumors (29.5%). Ninety-three patients (13.2%) had recurrent tumors and 97 (13.7%) died from their malignant disease. The robust prognostic value of FIGO stage and

**Funding:** The authors received no specific funding for this work.

**Competing interests:** The authors have declared that no competing interests exist.

lymph node status were confirmed. Other important survival predictors were histological grade and LVSI [overall survival: hazard ratio (HR) = 3.75, p < 0.001 and HR = 2.01, $p$ = 0.001; recurrence: HR = 2.49, $p$ = 0.004 and HR = 3.22, $p$ = 0.001, respectively). Disease-free ($p$ = 0.087) and overall survival ($p$ = 0.368) did not differ significantly between patients with stage II and III disease. These results indicate that prognostic role of cervical involvement should be explored further. This study reports the characteristics and outcomes of endometrial cancer in a large population from a single institution, with systematic surgical staging, a predominance of minimally invasive procedures, and well-documented outcomes. Prognostic factors in the present study population were generally similar to those in other countries, though our patients' tumors were larger than in studies elsewhere due to later diagnosis. Our unexpected finding of similar prognoses of stage II and III patients raises questions about the prognostic value of cervical involvement and possible differences between carcinomas originating in the lower uterine segment versus those originating in the body and fundus. The present findings can be used to guide public policies aimed at improving the diagnosis and treatment of endometrial cancer in Brazil and other similar countries.

## Introduction

Endometrial cancer is the sixth most common cancer in women worldwide and the second most common gynecological cancer in Brazil. Its incidence is increasing, in Brazil [1] and worldwide, with 382,096 new cases and 89,929 deaths from the disease recorded in 2018 [2]. By 2030, the incidence of endometrial cancer is projected to increase by 55% over the number of cases recorded in 2010 in the USA [3]. According to the Global Cancer Observatory's 2018 report [4], the numbers of new cases and deaths in Brazil were 9,105 and 2,472, respectively, with 15,091 new cases expected by 2040.

This increasing incidence is attributable mainly to cases of type I endometrial cancer caused by excess estrogen unopposed by progesterone. In Brazil, it can be attributed to factors such as increases in life expectancy, decreases in the number of pregnancies, and, especially, the increasing incidence of obesity, a strong risk factor for endometrial cancer [5]. Obesity has become a global epidemic, especially among women [6]. In 2017, an estimated 51.2% of Brazilian women were found to be overweight [7]. The prevalence of obesity among Brazilian women increased from 7.8% in the mid-1970s to 16.9% in 2008 and 24.4% in 2013, and the increase has been the most dramatic among low-income women [8–10].

Little information is available regarding the characteristics and outcomes of endometrial cancer in Brazil. The goal of this study was to describe 10 years of experience in the treatment of endometrial cancer at the Instituto do Cancer do Estado de Sao Paulo (ICESP), a public tertiary hospital and university-linked teaching center. The ICESP began these treatment programs in 2008 and receives patients from regions throughout Brazil.

## Methods

### Institutional approval

The protocol for this study was approved by the Scientific Committee of the Faculdade de Medicina da Universidade de Sao Paulo and by the Ethics Committee for Research Projects of the Hospital das Clınicas da Faculdade de Medicina da Universidade de Sao Paulo (Comissao de Ética para Análise de Pesquisa—CAPPesq) and Plataforma Brasil (CAAE060630194000

00065). It complies with the ethical precepts proposed by the legislation in force in Brazil R466/2012. The specific informed consent form for this work was waived by the aforementioned Ethics Committee because the study was retrospective, involving data from medical records, with no risks or benefits arising from the results, and with guarantees of full anonymity, including non-public sharing of data.

## Study cohort and statistical analysis

This retrospective study involved the analysis of data from 703 patients diagnosed with endometrial cancer who were treated at the ICESP between December 2008 and January 2018. Data were imported from the REDCap database. All diagnoses were based on the pathological study of surgical specimens by a team of pathologists at the ICESP Hospital following the guidelines of the College of American Pathologists [11]. Other data analyzed were age at diagnosis, ethnicity, body mass index, Karnofsky performance status (KPS), serum CA125, histological type and grade, lymphovascular space invasion (LVSI) (present or absent), tumor size, depth of myometrial infiltration (<50% or ≥50%), and surgical staging according to the 2009 International Federation of Gynecology and Obstetrics (FIGO) criteria [12]. The evaluated patients were divided into two age groups (≤50 years and >50 years); this cutoff was used because the average age of natural menopause in Brazil is 50 years old [13]. Additionally, parameters were compared between patients without functional disabilities (KPS 90 or 100) and patients with an objective loss of functional capacities (KPS 70 or lower). Outcomes were reported according to histological type, surgical FIGO stage, LVSI, and lymph node metastasis. Tumor size was dichotomized in two categories according two different cutoffs for analysis: 2 cm and 4 cm. The concordance between histological biopsy and surgical specimen findings was analyzed. Frequencies and percentages were calculated for qualitative variables, whereas medians and ranges were determined for quantitative variables. Survival curves were constructed by the Kaplan-Meier method and categories of cases were compared by log-rank testing. Multivariate Cox proportional hazard regression modeling with backward stepwise selection of variables was used to analyze other putative factors (beyond FIGO stage and lymph node status) that may be associated with overall survival and disease-free survival. Univariate and multivariate hazard ratios (HRs) were calculated together with 95% confidence intervals (95% CIs). Variables that were found to have $p$ values < 0.10 in the univariate analyses were selected for multivariate model analysis. For all hypothesis testing, a significance level of 5% was applied. The analyses were performed in SPSS v.25 for Windows software.

## Results

In the 10-year study period, 703 histologically confirmed cases of endometrial cancer were treated primarily with surgery at our institution. The clinicopathological features of the study sample are shown in Table 1. The following comorbidities were present: morbid obesity (body mass index > 35 kg/m$^2$) (12.7%), diabetes (33.2%), hypertension (62.9%), breast cancer (4.1%), and other cancers (6.1%). Tumor size ranged from 0 cm to 16.5 cm (median, 4.0 cm). Most tumors were low-grade endometrioid malignancies. High-grade tumors included endometrioid G3 and non-endometrioid histological types, with the latter accounting for 19.3% of the cases. The incidence of grade 3 tumors was similar in the first and last 5-year halves of the study period (6.0% *vs*. 8.4%).

The median time between the reporting of the first symptom (abnormal uterine bleeding) at a primary health assistance unit and pathological diagnosis was 9.0 months, and that between diagnosis and treatment at the ICESP was 4.0 months. There was histological concordance between biopsy-based and surgical specimen-based diagnoses in 411/703 cases (58.4%).

**Table 1. Characteristics of the study population.**

| Characteristic | Categories | N/median[a] | Range | % |
|---|---|---|---|---|
| **Age** | | 63[a] | 27–93 | |
| | ≤ 50 years | 42 | | 6.4 |
| | > 50 years | 661 | | 94 |
| **Body mass index** | | 31.1[a] | 16–58.9 | |
| **Race** | Caucasian | 550 | | 78.2 |
| | Black | 62 | | 8.8 |
| | Others | 91 | | 13 |
| **Histologic type** | Endometrioid G1/G2 | 468 | | 66.5 |
| | Endometrioid G3 | 99 | | 14.1 |
| | Serous | 93 | | 13.2 |
| | Clear cell | 28 | | 4 |
| | Others | 15 | | 2.1 |
| **FIGO (2009) stage** | IA | 298 | | 42.4 |
| | IB | 151 | | 21.5 |
| | II | 50 | | 7.1 |
| | III | 159 | | 22.6 |
| | IV | 45 | | 15 |
| **KPS** | ≥90 | 533 | | 75.8 |
| | ≤70 | 53 | | 7.5 |
| **Surgery type** | Laparotomy | 182 | | 25.9 |
| | Laparoscopic | 480 | | 68.3 |
| | Robotic | 43 | | 6.1 |
| | Vaginal | 2 | | 0.3 |
| **Myometrial invasion** | ≥50% | 343 | | 48.8 |
| | <50% | 343 | | 48.8 |
| **Tumor size** | <2 cm | 96 | | 13.7 |
| | 2–4 cm | 326 | | 46.4 |
| | >4 cm | 275 | | 39.1 |
| **Lymph node dissection** | | 551 | | 78.4 |
| | Pelvic only | 99 | | 14.1 |
| | Pelvic and para-aortic | 432 | | 61.5 |
| | Para-aortic only | 6 | | 0.9 |
| **Median no. lymph nodes recovered** | Pelvic | 15 | 1–61 | |
| | Para-aortic | 8 | 1–54 | |
| **Patients with positive lymph nodes** | | 125 | | 17.7 |
| | | | | |
| **LVSI+** | | 208 | | 29.5 |
| **Median surgical time (min)** | | 240 | 57–775 | |
| | Laparoscopic | 256 | 70–681 | |
| | Robotic | 320 | 170–643 | |
| | Laparotomy | 180 | 57–775 | |
| **Median hospital stay (days)** | | 3 | 2–57 | |
| **Adjuvant treatment** | | 414 | | 58.9 |
| | Chemotherapy | 241 | | 34.2 |
| | External beam radiation therapy | 186 | | 26.4 |
| | Brachytherapy | 174 | | 24.7 |
| **Recurrence** | | 93 | | 13.2 |

*(Continued)*

**Table 1.** (Continued)

| Characteristic | Categories | N/median[a] | Range | % |
|---|---|---|---|---|
| | Pelvic only | 16 | | 2.3 |
| | Distant only | 69 | | 9.8 |
| | Pelvic and distant | 12 | | 1.7 |
| **Death** | | 120 | | 17 |

When tumors were grouped simply as low or high grade, the concordance rate reached 532/703 cases (75.7%).

Before surgery, 583 patients underwent CA125 blood testing, which yielded median values of 21.9 (range, 3–2293) U/mL overall, 15.4 (3–2220) U/mL among patients with stage I tumors, and 37.2 (5.3–2293) U/mL among those with stages III or IV tumors. High and low CA125 level groups were defined by a median cut-off value of 21.9 U/ml. The HR for recurrence was 1.58 (95% CI 1.01–2.49; $p = 0.047$). For overall survival, the HR was 1.86 (95% CI 1.23–2.81; $p = 0.003$).

The number of minimally invasive surgeries increased from 36.4% in the first 2 years of hospital operation (2008 and 2009) to 90.3% in 2017. Lymph node dissection was performed in a total of 551 of the 703 patients (78.4%), systematic pelvic and para-aortic lymphadenectomies were performed in 432 cases (61.5%), pelvic lymphadenectomy alone was performed in 99 cases (14.1%), and para-aortic lymphadenectomy was performed alone in 6 cases (0.9%). The mean numbers (standard deviations) of pelvic and para-aortic lymph nodes dissected were 15 (10.7) and 8 (9.2), respectively. Lymph nodes were positive in 124 cases (17.7%). The median hospital stay was 3 (range, 2–57) days.

The median length of surgery was 240 (range, 57–775) min [laparoscopic, 256.5 (70–681) min; robotic, 320 (170–643) min; laparotomy, 180 (57–775) min]. The percentages of patients with high performance status (KPS > 70) according to surgery type were 92.9% for minimally invasive surgery cases and 86.3% for laparotomy cases. Pelvic and para-aortic lymphadenectomies were performed in 70% of patients who underwent minimally invasive surgery and in 37.9% of those who underwent laparotomy.

Thirty-nine (5.5%) procedures were converted to open surgery, due mainly to technical difficulties encountered during the initial surgeries [$n = 24$ (61.5%)], followed by the presence of vascular lesions [$n = 9$ (23%)], surgical device malfunctioning [$n = 4$ (10.3%)], and anesthesia-related issues [$n = 2$ (5.1%)]. In 15 minimally invasive surgeries (2.9%), small laparotomies were performed for surgical specimen removal due to large specimen size.

After surgery, 414 patients (58.9%) received adjuvant treatment, 241 (58.2%) received chemotherapy, 186 (44.9%) received external beam radiation therapy, and 174 (42%) received brachytherapy. Twenty-two patients (3.1%) received neoadjuvant treatment because they had advanced unresectable tumors.

During the follow-up period, recurrence was detected in 93/703 patients (13.2%). The recurrence rate did not differ significantly among patients with stage II and stage III disease (20% and 25.8%, respectively). Local recurrence in the vagina was diagnosed in 21 recurrence cases (22.6%), distributed as follow: patients with initial FIGO stages IA ($n = 4$), IB ($n = 3$), II ($n = 4$), IIIB ($n = 3$), IIIC1 ($n = 4$), IVA ($n = 1$), and IVB ($n = 2$). Seventy percent of patients with local vaginal recurrence had received adjuvant therapy. Inguinal lymph node recurrence was observed in 3 patients (3.2%) with initial stages of IA, IB, and IIIC2. Distant lymph-node metastasis occurred in 28 of the 93 patients with recurrence (30.1%) [initial FIGO stages IA ($n = 2$), IB ($n = 4$), II ($n = 6$), IIIA ($n = 1$), IIIC1($n = 5$), IIIC2 ($n = 4$), IVA ($n = 1$), and IVB

($n$ = 5)]. Recurrence was detected in 3.4% and 7.9% of patients with stages IA and IB, disease, respectively. In patients with advanced stage disease, recurrence with <50% and ≥50% myometrial infiltration was detected in 21.4% and 29.1% of cases, respectively. Recurrence was diagnosed as peritoneal carcinomatosis in 22/93 (23.7%) cases, hepatic metastasis in 7/93 (7.5%) cases, lung metastasis in 22/93 (23.7%) cases, bone metastasis in 5/93 (5.4%) cases, and brain metastasis in 2/93 (2.2%) cases.

Of the 703 patients in the study sample, 97 (13.7%) died during the follow-up period (Fig 1A). Higher overall and disease-free survival rates were associated with endometrioid histological types (Fig 1B), FIGO stage I (Fig 1C), negative lymph nodes (Fig 2A), and absence of LVSI (Fig 2B). Disease-free and overall survival did not differ between patients with stage II and stage III disease (Fig 3)

In the univariate analysis, the following parameters were identified as prognostic factors: tumor histology, histological grade, LVSI, tumor size, and myometrial infiltration. All variables had HRs > 2, except for tumor size ≥ 4 cm for recurrence. In the adjusted analysis, histological grade and LVSI remained independent prognostic factors (Table 2). High histological grade presented the greatest risk for both death (HR = 3.75, $p$ < 0.001) and recurrence (HR = 2.49, $p$ = 0.004).

## Discussion

Endometrial cancer occurs mainly in postmenopausal women and is linked to obesity; the ages and body mass index values of our patients reflect this pattern. In addition, the relationship between hypertension and endometrial cancer, which has been recognized for decades [14–17], was apparent in this study. In a Danish population study [18], 56.6% of patients with endometrial cancer had no comorbidity. Only 12.5% of the patients in our study sample had no comorbidity.

The present study confirmed the importance of classical prognostic factors, namely histological type and grade, LVSI, tumor size, and myometrial invasion.

Histological grade was associated with the greatest risks of death and recurrence in this study. The European Society for Medical Oncology (ESMO)–European Society of Gynaecological Oncology (ESGO)–European Society for Radiotherapy & Oncology (ESTRO) consensus conference statement assigns intermediate risk to grade III endometrioid carcinoma, even in cases of FIGO stage IA tumors without LVSI [19]. Mang et al. [20] observed an increase in grade 3 endometrioid carcinoma diagnoses from 18% to 32% between 2006 and 2014 in a sample of 2,611 patients. We observed no such increase.

LVSI is considered almost unanimously to be predictive of a poor prognosis [14–16], and this supposition was confirmed in this study in which LVSI increased the risk of recurrence threefold. The ESMO-ESGO-ESTRO consensus assigns intermediate to high risk to early-stage carcinomas [19]. This prognostic information becomes very important for decision-making about adjuvant treatment in patients with early-stage disease [21, 22].

LVSI has been recognized as a powerful predictor of lymph node involvement [23]. However, the usefulness of this information in deciding whether or not to perform a lymphadenectomy depends on the histology of the surgical specimen, which is limited to frozen section examination during operations. It would be interesting to identify reliable estimates of risk of lymph node involvement based on biopsy findings. In a previous study with 47 consecutive patients who had complete surgical staging, we found LVSI in 63.4% of cases, and L1CAM positivity in 17% of the cases. All L1CAM-positive cases had LVSI, suggesting a potential role of this molecule at the time of diagnostic biopsy, particularly in identifying patients in whom lymphadenectomy is crucial [24].

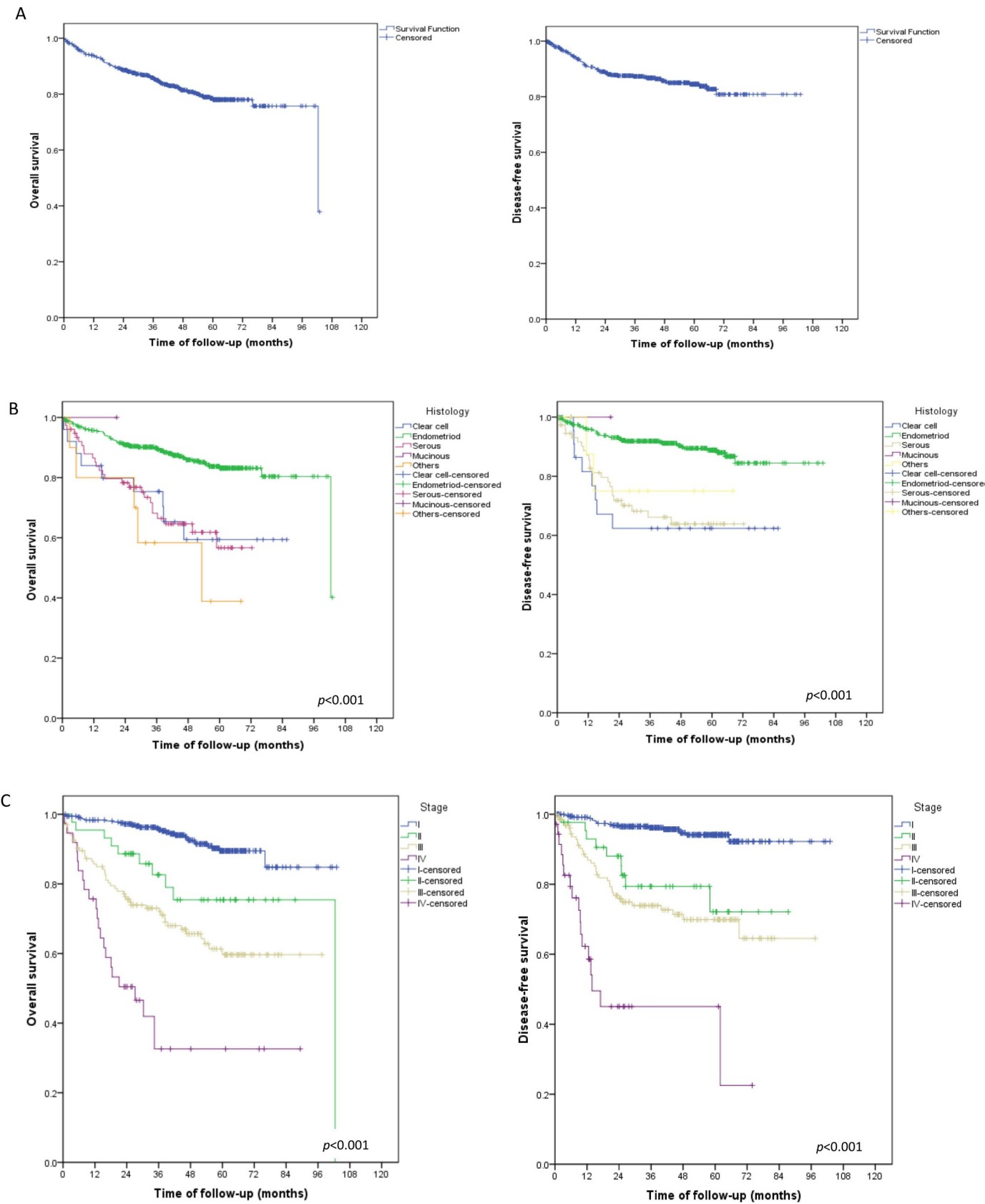

**Fig 1. Overall and disease-free survival for the whole cohort and by histological subtype and FIGO stage.** (A) Survival rates for entire cohort of patients with endometrial cancer.(B) Survival rates by histological subtype. (C) Survival rates by FIGO stage.

Tumors were larger in our study population than in patients with endometrial carcinoma in studies from developed countries. In a series of 703 cases reported by AlHilli *et al.* [25], 29.8% of tumors were ≤2 cm in size whereas in our cohort only 13.7% of tumors were ≤2 cm. This difference may be attributable to delayed care and resolution of cases in our study population. The intervals between initial symptom reporting, diagnosis, and treatment support this assumption. Delayed diagnosis and treatment initiation are strongly related to the prognosis of patients with endometrial carcinoma, and longer wait times from diagnosis to surgery have a negative impact on survival [26]. Outpatient endometrial biopsy is not widespread in Brazil; most gynecologists prefer to refer patients with abnormal uterine bleeding to hysteroscopy

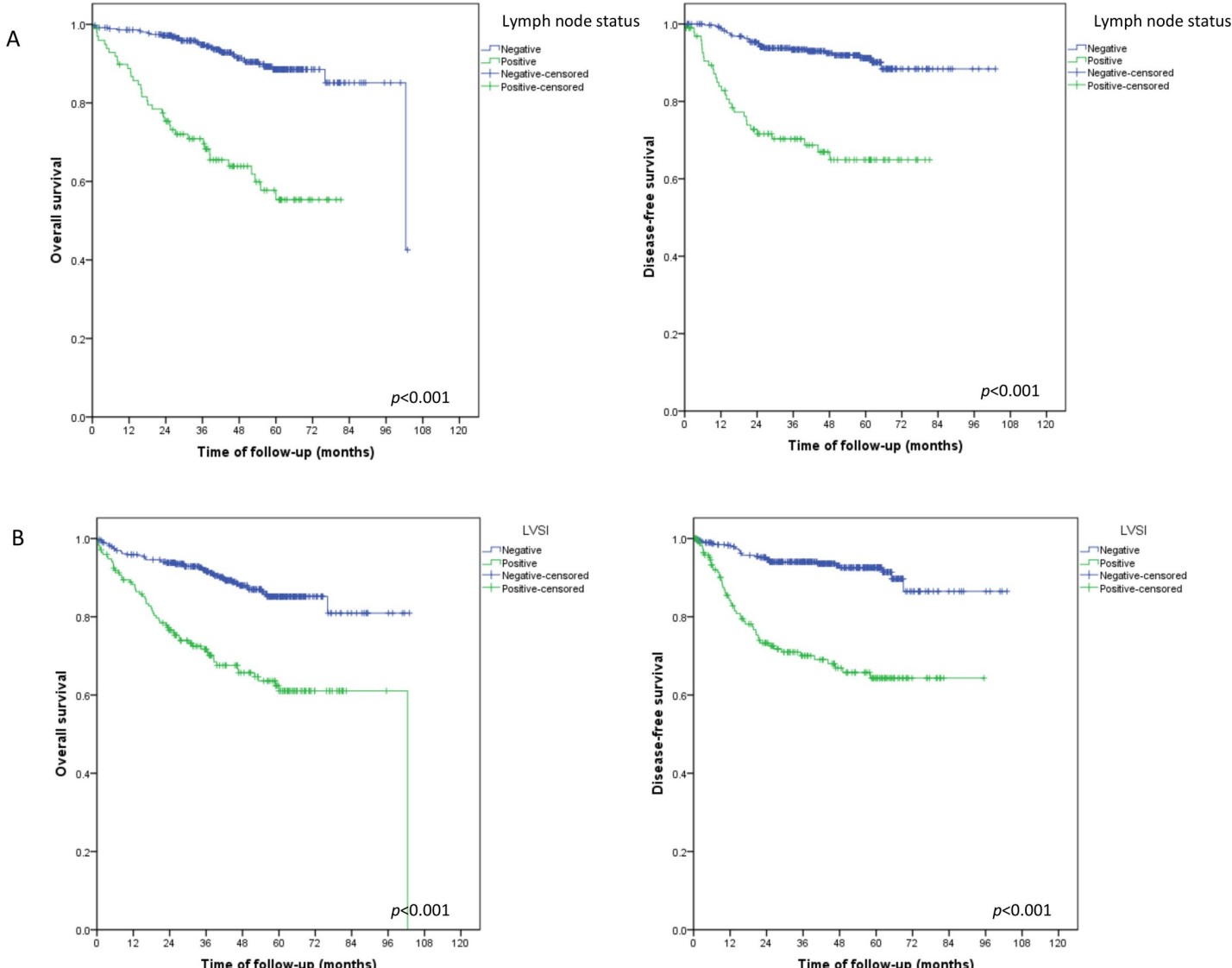

**Fig 2. Overall and disease-free survival by lymph node status.** (A) lymph node status. (B)lymph vascular space invasion (LVSI).

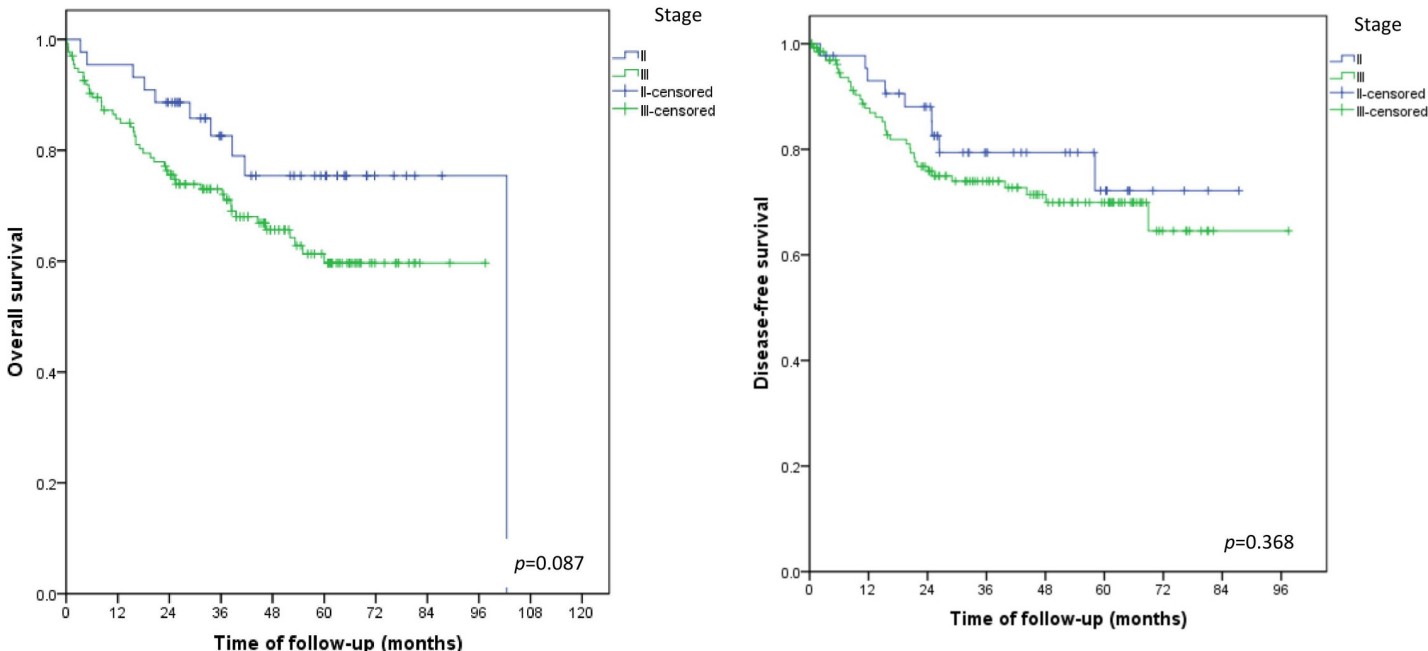

**Fig 3. Overall and disease-free survival in FIGO stage II and III.**

services. In our opinion, this practice may be a major factor contributing to delayed diagnosis. We have thus been encouraging our colleagues to spread the practice of outpatient endometrial biopsy.

The type of surgery used to treat endometrial carcinomas changed considerably at our institution over the study period, with a marked increase in the performance of minimally invasive (traditional laparoscopic and robotic) surgeries [27–29]. Currently, laparoscopy is the treatment of choice, employed in >90% of cases. Robotic surgery is performed primarily within the bounds of research projects due to its high cost in Brazil, especially in a public service setting

**Table 2. Factors associated with overall survival and disease-free survival in patients diagnosed with endometrial cancer.**

| Parameters associated with overall survival | | Univariate | | Multivariate | |
|---|---|---|---|---|---|
| | | HR (95% CI) | *p* | HR (95% CI) | *p* |
| Histologic type | | 3.1 (2.01–4.6) | <0.0001 | | |
| Histologic grade (high *vs.* low) | | 4.94 (3.30–7.40) | <0.0001 | 3.75 (2.42–5.82) | <0.001 |
| LVSI | | 3.26 (2.21–4.80) | <0.0001 | 2.01 (1.32–3.06) | 0.001 |
| Tumor size | 2 cm | 2.46 (1.13–5.32) | 0.023 | | |
| | 4 cm | 2.25 (1.43–3.52) | <0.001 | | |
| Myometrial invasion | | 3.20 (1.91–5.35) | <0.0001 | | |
| **Parameters associated with disease free survival** | | | | | |
| Histologic type | | 3.82 (2.44–5.98) | 0.001 | | |
| Histologic grade | | 5.24 (3.30–8.32) | 0.001 | 2.49 (1.33–4.64) | 0.004 |
| LVSI | | 5.02 (3.16–7.97) | 0.001 | 3.22 (1.95–5.32) | 0.001 |
| Tumor size | 2 cm | 2.03 (0.88–4.70) | 0.099 | | |
| | 4 cm | 1.66 (1.00–2.74) | 0.05 | | |
| Myometrial invasion | | 2.78 (1.81–4.29) | 0.001 | | |

[27]. Complete surgical staging with pelvic and para-aortic lymph node dissection was performed in most of our patients, but lymph nodes were positive in a relatively small percentage of cases. Other prognostic factors, such as molecular profile factors, may aid in the identification of patients at low risk of lymph node involvement who may be well-suited for potential treatment in community hospitals.

The rate of local recurrence among patients with early (stage I or II) disease was about 1.2% in our sample, and most patients with vaginal recurrence had received adjuvant radiotherapy. Recently, Francis *et al.* [30] reported a relatively high vaginal recurrence rate (3.7%), probably because only 25% of their patients received adjuvant radiotherapy. An unexpected finding of our study was that patients with stage II and stage III tumors had similar recurrence rates as well as similar disease-free ($p = 0.087$) and overall survival ($p = 0.368$) outcomes. It is expected that prognoses would be better for patients with tumors invading only the stroma of the cervix, with no extension beyond the uterus (stage II), than for those with invasion of the uterine serosa, parametrium, or lymph nodes (stage III). For endometrial cancer, FIGO stage II does not differentiate between tumors originating in the body of the uterus and extending contiguously to the cervix from those originating in the lower uterine segment and extending to the cervix, despite the quite different behaviors of these two disease presentations. Lymphatic drainage differs between the lower and upper segments of the uterus. Lower uterine segment involvement (LUSI) correlates with more LVSI and other poor prognostic factors, such as lymph node metastasis, uterine serosal involvement, and deep myometrial invasion [31]. Cokmez and Yilmaz reported LUSI in 49 (19.4%) of 253 cases of endometrioid endometrial carcinoma [32]. Patients with LUSI presented more LVSI (36.7% *vs.* 11.8%, $p < 0.05$) and more lymph node metastasis (18.4% *vs.* 4.9%, $p < 0.05$) than patients without LUSI [32]. Pathologists should report LUSI because this information could be used as a staging criterion for endometrial carcinoma in the future.

This study has several strengths. First, it presents the largest case series of Brazilian patients with endometrial cancer from a single tertiary institution with a uniform surgical and pathology team, following rigid guidelines, reported thus far. The tumors in all of the cases were submitted to systematic surgical staging. Additionally, the patients underwent predominantly minimally invasive procedures, and every patient in the series had a well-documented outcome. Notably, our finding of similar prognoses among stages II and III cases should prompt an important discussion about the significance of cervical involvement. Furthermore, our data raise questions about the importance of differentiating between tumors originating in the lower uterine segment from those originating in the uterine body or fundus. That is, it should be considered whether lower uterine segment carcinomas and endometrial carcinomas should be treated as distinct entities.

Our study had some weaknesses. First, there may be biases in our data associated with the study being retrospective. Secondly, the study period encompassed a relatively long primary treatment period (10 years), during which there was a learning curve for minimally invasive surgery, a reduction in the number of open operations, and changes in the standards of care for adjuvant treatments. Lastly, we focused on the analysis of classical variables, without examining molecular factors. Notwithstanding, the large number of cases subjected to uniform analyses in this study allowed us to delineate the presentation of endometrial cancer in a Brazilian patient population and our results may thus provide information that is useful for health policy development in Brazil and in other countries with characteristics similar to Brazil.

## Conclusion

Our study confirms the importance of the classical prognostic factors of histological type and LVSI as strong predictors of poor prognosis in patients diagnosed with endometrial cancer. The present findings of similar disease-free and overall survival outcomes between patients with stage II versus stage III disease indicate that the prognostic value of cervical involvement and tumor origination in the lower uterine segment should be explored further. Although most of the cases in this series had early-stage diagnoses, we observed tumors that were larger in size than are commonly observed in studies performed in more developed countries, presumably due to later diagnosis. The findings of this study can be used to guide public policies aimed at improving the timeliness of endometrial cancer diagnosis and treatment in Brazil and in other low and medium developing countries.

## Supporting information

**S1 File. Dataset.**
(XLSX)

## Author Contributions

**Conceptualization:** Cristina Anton, Jesus Paula Carvalho.

**Data curation:** Eric Mayerhoff, Maria del Pilar Esteves Diz, Daniela de Freitas, Heloisa de Andrade Carvalho, João Paulo Mancusi de Carvalho, Alexandre Silva e Silva, Maria Luiza Nogueira Dias Genta, Rafael Calil Salim, Andrea Aranha.

**Formal analysis:** Cristina Anton, Rossana Veronica Mendoza Lopez, Jesus Paula Carvalho.

**Methodology:** Cristina Anton, Jesus Paula Carvalho.

**Project administration:** Cristina Anton.

**Supervision:** Edmund Chada Baracat, Jesus Paula Carvalho.

**Writing – original draft:** Cristina Anton, Rodolpho Truffa Kleine, Jesus Paula Carvalho.

**Writing – review & editing:** Cristina Anton, Maria Luiza Nogueira Dias Genta, André Lopes de Faria e Silva, Filomena Marino Carvalho, Jesus Paula Carvalho.

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
