## [Decision Letter · Decision Letter 0]

6 Jan 2020

PONE-D-19-32782

Ten years of experience with endometrial cancer treatment in a single Brazilian institution: Patient characteristics and outcomes

PLOS ONE

Dear Dr. Cristina Anton,

Thank you for submitting your manuscript to PLOS ONE. After careful consideration, we feel that it has merit but does not fully meet PLOS ONE’s publication criteria as it currently stands. Therefore, we invite you to submit a revised version of the manuscript that addresses the points raised during the review process.

ACADEMIC EDITOR: The manuscript might be significant for public health strategy in Brazil. However, the analysis of prognostic factors in Table 2 only by a univariate analysis seems inadequate. Please consider a multivariate analysis for all the factors before you draw a scientific conclusion. Also, there is no limitation in this study mentioned. Please provide a paragraph of the potential limitations in the present analysis. 

We would appreciate receiving your revised manuscript by Feb 20 2020 11:59PM. To enhance the reproducibility of your results, we recommend that if applicable you deposit your laboratory protocols in protocols.io, where a protocol can be assigned its own identifier (DOI) such that it can be cited independently in the future. For instructions see: http://journals.plos.org/plosone/s/submission-guidelines#loc-laboratory-protocols

We look forward to receiving your revised manuscript.

Kind regards,

Jason Chia-Hsun Hsieh, M.D. Ph.D

Academic Editor

PLOS ONE

2. Thank you for including your ethics statement in the manuscript: 'This retrospective study (institutional review board approval no. 457913–2018) involved the analysis of data from 703 patients diagnosed with endometrial cancer attended at the ICESP between December 2008 and January 2018.'

a) Please amend your current ethics statement to include the full name of the ethics committee/institutional review board(s) that approved your specific study and confirm that your named institutional review board or ethics committee specifically approved this study.

3. In the ethics statement in the manuscript and in the online submission form, please provide additional information about the patient records/samples used in your retrospective study. Specifically, please ensure that you have discussed whether all data/samples were fully anonymized before you accessed them and/or whether the IRB or ethics committee waived the requirement for informed consent. If patients provided informed written consent to have data/samples from their medical records used in research, please include this information.

4. We noticed you have some minor occurrence(s) of overlapping text with the following previous publication(s), which needs to be addressed:

https://doi.org/10.3802/jgo.2018.29.e100

In your revision ensure you cite all your sources (including your own works), and quote or rephrase any duplicated text outside the Methods section. Further consideration is dependent on these concerns being addressed.

Additional Editor Comments (if provided):

The manuscript might be significant for public health strategy in Brazil. However, the analysis of prognostic factors in Table 2 only by a univariate analysis seems inadequate. Please consider a multivariate analysis for all the factors before you draw a scientific conclusion. Also, there is no limitation in this study mentioned. Also, please provide a paragraph of the potential limitations in the present analysis.

Reviewers' comments:

Reviewer's Responses to Questions

**Comments to the Author**

1. Is the manuscript technically sound, and do the data support the conclusions?

Reviewer #1: Partly

Reviewer #2: Yes

2. Has the statistical analysis been performed appropriately and rigorously? 

Reviewer #1: No

Reviewer #2: Yes

3. Have the authors made all data underlying the findings in their manuscript fully available?

Reviewer #1: No

Reviewer #2: Yes

4. Is the manuscript presented in an intelligible fashion and written in standard English?

Reviewer #1: No

Reviewer #2: No

5. Review Comments to the Author

Reviewer #1: Im very appreciated to have reviewing this manuscript. The purpose of retrospective study is analyzed and evaluated the characteristics and prognostic factors of 703 patients with endometrial cancer treated at a single center over ten years, The data showed in this study was interesting and meaningful, but I have some comments and suggestions:

1. Firstly, please submit as a regular and standardized format of Plos One, it will help for the whole system and structure of this article;

2. Weaknesses are retrospective study and no something new point. It is better to describe more reasons to accept the authors' hypothesis.

3. In Table 1, the what is the foundation of your claim to stratified “Age” as 53 years, and “KPS stratified as 70 and 90”, and so on in the Table1. The methods are not mention in the statistical analysis.

4. This study is a valuable study but I think it is somewhat insufficient to publish in this journal.

Reviewer #2: In this study, the authors analyze the characteristics, prognostic factors and outcomes of patients with endometrial cancer treated at a single tertiary Brazilian institution over a 10-year period. It was finally concluded that histological type and LVSI were strong prognostic predictors in this population. This study reports the characteristics and outcomes of endometrial cancer in a large population from a single institution. Its findings may be used to guide public policies of endometrial cancer in Brazil.

In my opinion, the methodology of this manuscript is correct and the statistical is appropriate. However, what is the innovation of the study? The prognostic factors related to endometrial cancer mainly include clinical stage, histological type, LN metastatic status, LVSI, etc. Why is the conclusion of this study only related to histological type and LVSI? It is recommended that the author give interpretation and discussion and highlight the novelty of the study.

Moreover, the manuscript needs substantial revision and improvement in English writing which are critical to the understanding of the research. Here are some comments；

1) “The disease-free and overall survival did not differ significantly between patients with stages II and III disease.”

Please give the P values in each of the situation.

2) “Tumor size ranged from 0 to 16.5 cm (median 4cm Most tumors were endometrioid low grade.”

There missed a “)” after “4cm”.

3) “Before surgery, 583 patients underwent CA125 testing. The median values were 21.9 (range 3–2293) U/mL overall, 15.4 (3–2220) U/mL among patients with stage I tumors, and 37.2 (5.3–2293) U/mL among those with stages III and IV tumors.”

Please analysis whether there is a survival difference between high CA125 group and low CA125 group (the author can use 21.9U/ml as a cutoff value).

4) Please describe the method of calculating the adjusted hazard ratios (HR).

6. PLOS authors have the option to publish the peer review history of their article (what does this mean?). If published, this will include your full peer review and any attached files.

Reviewer #1: No

Reviewer #2: No

---

## [Author Response · Author response to Decision Letter 0]

7 Feb 2020

PONE-D-19-32782

Ten years of experience with endometrial cancer treatment in a single Brazilian institution: Patient characteristics and outcomes

R.: We thank the editor and the reviewers for the careful analysis of our work. Please find below our answers and we are available for further clarification.

R.: All the format alterations according PLoS style were done and we chose to highlight (in yellow) only text modifications and not formatting changes.

2. Thank you for including your ethics statement in the manuscript: 'This retrospective study (institutional review board approval no. 457913–2018) involved the analysis of data from 703 patients diagnosed with endometrial cancer attended at the ICESP between December 2008 and January 2018.'

a) Please amend your current ethics statement to include the full name of the ethics committee/institutional review board(s) that approved your specific study and confirm that your named institutional review board or ethics committee specifically approved this study.

b) Once you have amended this/these statement(s) in the Methods section of the manuscript, please add the same text to the “Ethics Statement” field of the submission form (via “Edit Submission”).3. In the ethics statement in the manuscript and in the online submission form, please provide additional information about the patient records/samples used in your retrospective study. Specifically, please ensure that you have discussed whether all data/samples were fully anonymized before you accessed them and/or whether the IRB or ethics committee waived the requirement for informed consent. If patients provided informed written consent to have data/samples from their medical records used in research, please include this information. 

R.: We included the statement in the Methods statement, highlighted in yellow.

4. We noticed you have some minor occurrence(s) of overlapping text with the following previous publication(s), which needs to be addressed:

https://doi.org/10.3802/jgo.2018.29.e100

In your revision ensure you cite all your sources (including your own works), and quote or rephrase any duplicated text outside the Methods section. Further consideration is dependent on these concerns being addressed. 

R.: Sorry, but we couldn’t identify the overlapping text. If you can help us by pointing out the sections with a problem more accurately, we appreciate it. 

Additional Editor Comments (if provided):

The manuscript might be significant for public health strategy in Brazil. However, the analysis of prognostic factors in Table 2 only by a univariate analysis seems inadequate. Please consider a multivariate analysis for all the factors before you draw a scientific conclusion. Also, there is no limitation in this study mentioned.Also, please provide a paragraph of the potential limitations in the present analysis.

R.: Thanks for the comments. The multivariate analysis was done and it is in 4th and 5th columns of Table 2. The description of the statistical method is highlighted in blue in the Method section. Potential limitations as strengths were highlighted in green, in the Discussion section. 

 

Reviewers' comments:

Reviewer #1: Im very appreciated to have reviewing this manuscript. The purpose of retrospective study is analyzed and evaluated the characteristics and prognostic factors of 703 patients with endometrial cancer treated at a single center over ten years, The data showed in this study was interesting and meaningful, but I have some comments and suggestions:

R.: Thanks for the comments and useful suggestions.

1. Firstly, please submit as a regular and standardized format of Plos One, it will help for the whole system and structure of this article;

R.: All the format alterations according PLoS style were done. We chose to highlight (in yellow) only text modifications and not formatting changes.

2. Weaknesses are retrospective study and no something new point. It is better to describe more reasons to accept the authors' hypothesis. 

R.: Thanks for the comment. We agree and we have provided a more detailed description of the strengths and weaknesses in the discussion section (highlighted in green).

3. In Table 1, the what is the foundation of your claim to stratified “Age” as 53 years, and “KPS stratified as 70 and 90”, and so on in the Table1. The methods are not mention in the statistical analysis.

R.: The 50y cut-off was selected because is the average age of natural menopause in Brazil, so we could have an idea of premenopausal incidence. We explained better in the Method section. 

The KPS 90 or 100 was compared to 70 or less because we want to identify the proportion of patients without disabilities vs. those with significant symptoms, excluding the cases doubtful. We justified in the Method section (highlighted in green).

4. This study is a valuable study but I think it is somewhat insufficient to publish in this journal.

R.: We hope that after the changes, mostly motivated by your comments, the paper become more attractive.

Reviewer #2: 

In this study, the authors analyze the characteristics, prognostic factors and outcomes of patients with endometrial cancer treated at a single tertiary Brazilian institution over a 10-year period. It was finally concluded that histological type and LVSI were strong prognostic predictors in this population. This study reports the characteristics and outcomes of endometrial cancer in a large population from a single institution. Its findings may be used to guide public policies of endometrial cancer in Brazil.

In my opinion, the methodology of this manuscript is correct and the statistical is appropriate. However, what is the innovation of the study? The prognostic factors related to endometrial cancer mainly include clinical stage, histological type, LN metastatic status, LVSI, etc. Why is the conclusion of this study only related to histological type and LVSI? It is recommended that the author give interpretation and discussion and highlight the novelty of the study.

R.: Thanks for the comments. We provided some changes, particularly in discussion, including weaknesses and strengths. Furthermore, we emphasized our finding related to similar prognosis of stages II and III (highlighted in green in the Discussion section and in the abstract). Besides, this study exposes an important problem, common to less developed countries, which is late diagnosis and encourages a changing in the approach of diagnosis (office biopsy instead of hysteroscopy).

Moreover, the manuscript needs substantial revision and improvement in English writing which are critical to the understanding of the research. 

R.: We sent this version of the manuscript to revision by a professional edition service (Write Science Right; see certificate of edit) and they made a lot of changes. These changes are highlighted in pink. Please, let me know if it’s OK. 

Here are some comments；

R.: Thanks for observations.

1) “The disease-free and overall survival did not differ significantly between patients with stages II and III disease.”

Please give the P values in each of the situation.

R.: Done. Highlighted in blue. 

2) “Tumor size ranged from 0 to 16.5 cm (median 4cm Most tumors were endometrioid low grade.”

There missed a “)” after “4cm”. 

R.: Thanks! Highlighted in blue

3) “Before surgery, 583 patients underwent CA125 testing. The median values were 21.9 (range 3–2293) U/mL overall, 15.4 (3–2220) U/mL among patients with stage I tumors, and 37.2 (5.3–2293) U/mL among those with stages III and IV tumors.”

Please analysis whether there is a survival difference between high CA125 group and low CA125 group (the author can use 21.9U/ml as a cutoff value).

R.: Thanks for the suggestion. We included HR for DFS and OS using the cut-off of median (21.9U/ml) (highlighted in blue). 

4) Please describe the method of calculating the adjusted hazard ratios (HR).

R.: The method was described in the inserted text highlighted in blue in the last paragraph of the Study cohort and statistical analysis

---

## [Editor Report · Decision Letter 1]

10 Feb 2020

Ten years of experience with endometrial cancer treatment in a single Brazilian institution: Patient characteristics and outcomes

PONE-D-19-32782R1

Dear Dr. Anton,

We are pleased to inform you that your manuscript has been judged scientifically suitable for publication and will be formally accepted for publication once it complies with all outstanding technical requirements.

With kind regards,

Jason Chia-Hsun Hsieh, M.D. Ph.D

Academic Editor

PLOS ONE

Additional Editor Comments (optional):

All the questions were answered and revised adequately.
---

## [Editor Report · Acceptance letter]

18 Feb 2020

PONE-D-19-32782R1 

Ten years of experience with endometrial cancer treatment in a single Brazilian institution: Patient characteristics and outcomes 

Dear Dr. Anton:

I am pleased to inform you that your manuscript has been deemed suitable for publication in PLOS ONE. Congratulations! Your manuscript is now with our production department. 

With kind regards,

on behalf of

Dr. Jason Chia-Hsun Hsieh 

Academic Editor

PLOS ONE